# Boosting Lightweight Sentence Embeddings with Knowledge Transfer from Advanced Models: A Model-Agnostic Approach

**Kadir Gunel** *,† and **Mehmet Fatih Amasyali** †

Department of Computer Engineering, Yildiz Technical University, 34220 Istanbul, Turkey; amasyali@yildiz.edu.tr

\* Correspondence: kadir.gunel@std.yildiz.edu.tr

† These authors contributed equally to this work.

**Abstract:** In this study, we investigate knowledge transfer between two distinct sentence embedding models: a computationally demanding, highly performant model and a lightweight model derived from word vector averaging. Our objective is to augment the representational power of the lightweight model by exploiting the sophisticated features of the robust model. Diverging from traditional knowledge distillation methods that align logits or hidden states of teacher and student models, our approach uses only the output sentence vectors of the teacher model for the alignment with the student models's word vector representations. We implement two minimization techniques for this purpose: *distance minimization* and *distance and perplexity minimization* Our methodology uses WMT datasets for training, and the enhanced embeddings are validated via Google's Analogy tasks and Meta's SentEval datasets. We found that our proposed models intriguingly retained and conveyed information in a model-specific fashion.

**Keywords:** knowledge transfer; knowledge distillation; sentence embeddings; neural networks; sBert; FastText

## 1. Introduction

Knowledge distillation is a favored method for transferring information from one domain to another. This technique involves a teacher model transmitting knowledge to a simpler student model. The differences between these models typically pertain to network parameters such as the number of layers and the size of weight matrices. The student model aims to replicate the representations taught by the teacher as closely as possible. However, training a student model through knowledge distillation can be time-consuming and resource-intensive. Furthermore, the resulting student model may still lack predictive capabilities, leading to inefficient energy usage.

Contrary to existing literature on knowledge distillation, this paper explores scenarios where applying distillation methods can be challenging. These situations may arise when only partial access to model parameters is available or when different models need to be redeveloped using the same programming language and deep learning framework (source code for all experiments and used data are shared publicly on https://github.com/kadir-gunel/BoostingSentenceEmbeds—accessed on 23 October 2023).

This paper primarily investigates knowledge transfer between two distinct model architectures: *FastText* and *sBert*. While the training language is English, these models differ completely in architecture, training data, and output vector dimensions. Notably, the end user doesn't have access to the model parameters of *FastText* pre-trained word embeddings. Moreover, both models are implemented using different programming languages.

From a model perspective, FastText excels in speed by leveraging its trained word embeddings to calculate sentence vectors through a *bag-of-words* approach, averaging the word vectors within a sentence. This methodology enables *FastText* to generate sentence embeddings with remarkable rapidity, facilitating highly efficient vector computations

that do not require the use of a GPU. However, this speed comes at the cost of accuracy, marking a significant trade-off. In contrast, *sBert* employs its transformer architecture to craft sentence embeddings. Although this slows down the generation process due to its computational intensity which requires use of a GPU, it results in better performance across various NLP tasks. This trade-off underscores *sBert*'s enhanced sentence representation capabilities, where it sacrifices speed for a higher degree of sentence representation, as compared to *FastText*.

Our aim is to design and evaluate models that simultaneously address word-level and sentence-level representations. We propose models that seek to bridge the gap between *FastText* word embeddings and *sBert* sentence vectors by producing more closely aligned sentence representations. To facilitate the information transfer, the WMT EN-ES dataset is used. Subsequently, the validation of the proposed methods is conducted through classification and semantic task similarity (STS) assessments from the SentEval datasets. The paper is organized as follows:

- An overview of the literature (Section 2);
- Proposed methodologies for constructing knowledge tranfer between *FastText* word embeddings and *sBert*: *distance minimization* and *distance and perplexity minimization* (Section 3);
- Used datasets for training and testing proposed models on various tasks (Section 4);
- The techniques and model configuration for training (Section 5);
- Represent the results of proposed models for knowledge transfer by evaluating obtained results on word and sentence level tasks (Section 6);
- Evaluate the obtained results, how the proposed models affect the word embedding space, and what the possible solutions are (Section 7).

## 2. Literature

The development of sentence vector representations has been a prolific area of study. Initially, methods centered on averaging pretrained word embeddings, as demonstrated by [1–3]. Building upon this foundation, Ref. [4] broadened the representation from word level to varying lengths of text, encompassing sentences, paragraphs, and documents. This advancement empirically outperformed the traditional bag-of-words model in various text classification and sentiment analysis tasks, showcasing the ability to surmount the limitations inherent in the conventional averaging approach. Further developments included techniques such as subtracting principal components [5] and adopting the skip-gram methodology [6], which [7] later refined for expedited training.

Concurrently, breakthroughs in neural architectures have revolutionized sentence processing [8]. A multitude of models have emerged to strike a balance between accuracy and computational efficiency, delivering proficient performance across diverse language tasks [9,10]. The advent of sophisticated models such as BERT has not only broadened the horizons of transfer learning but also catalyzed progress in the field of knowledge distillation within language understanding tasks [11–13]. These advancements have spurred researchers to explore diverse strategies for sentence embedding, with a notable direction being the adoption of siamese network architectures that harness BERT's robust capabilities. This innovative approach (*sBert*) not only leverages BERT's advanced deep learning features to produce embeddings that effectively capture complex semantic relationships within sentences but also accelerates the embedding generation process, offering a more efficient solution [14].

Emerging from this context, knowledge distillation as a concept has been rigorously explored as a strategic means to transfer the essence of complex models into more simplified counterparts [15]. Typically, these approaches involve models with identical architectures, utilizing logits (pre-softmax outputs) from a teacher model to guide the student model towards producing similar outputs. Refs. [16,17] have integrated these approaches for language-agnostic sentence embeddings. Notably, these methods do not attempt to transfer knowledge between different architectures.

As the utilization of large scale pre-trained models in NLP expands, the issue of managing these models within limited computational resources becomes increasingly evident. In response, Ref. [18] introduced DistilBERT, a smaller, versatile language representation model that applies knowledge distillation during pre-training. This technique reduces the model's size while maintaining robust language comprehension, thus offering a practical solution for on-device calculations and an array of tasks.

Despite the increasing complexity of neural networks for NLP, as exemplified by Bert and GPT, some argue that simple neural networks can still compete if they are enhanced through knowledge distillation from these advanced models. Ref. [19] have demonstrated that with distillation, a single-layer BiLSTM and its siamese variant can perform on par with their more intricate counterparts, albeit with far fewer parameters and diminished inference time.

In a recent study, Ref. [20] presented a model that transfers knowledge from a transformer model to a convolutional model in the realm of image classification. Their approach aligns the student and teacher features in two projected feature spaces.

Our study introduces a novel approach in the field of sentence embeddings and knowledge transfer by employing a unique distillation method. Unlike previous works, we harness the distinct architectures of pre-trained *FastText* word embeddings with *sBert* sentence vector representations. This integration addresses the gap between robust sentence representation models and the need for computationally feasible solutions, charting a course for future research in the area.

## 3. Methodologies

*FastText* and *sBert* each creates their own embeddings, which results in distinct data points for the same sentence. *sBert* takes into account the context by considering word positions when generating sentence embeddings, while *FastText* uses a bag-of-words approach that loses information about the position of words. This fundamental difference leads to different orientations for representing sentences. However, both models aim to capture meanings from the input sentences, hence we expect some level of correlation between these vectors in their respective spaces (Figure 1). One way to enhance this correlation is by mapping these encodings onto a shared space and minimizing the distance between them. To achieve that, two methods are proposed:

1.  *Distance Minimization* aims to reduce the dissimilarity between *sBert* and *FastText* word embeddings by changing their representational direction.
2.  *Distance and Perplexity Minimization* aims to minimize the gap between *sBert* and *FastText* by checking the correctness of contextual information in addition to distance.

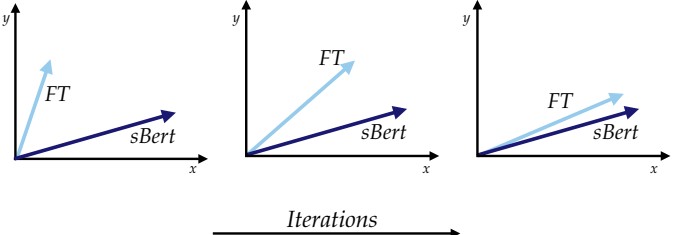

**Figure 1.** The three figures depict the distance minimization procedure. Two sentence vectors represented by *FastText* (light blue point) and *sBert* (dark blue point) sentence vectors belonging to the same sentence. The objective is to minimize the distance (angle) between the vectors by moving *FastText* sentence representations towards *sBert*.

Both approaches have the goal of improving the quality of word-level representations provided by *FastText*. To align representations with *sBert*, both proposed strategies employ representation distillation methods [18] that make use of only sentence embeddings generated by *sBert*. The primary aim is to ensure that the resulting embeddings remain consistent

with the benchmark *sBert* model. Once this distillation process is completed, the updated *FastText* word embeddings will be exclusively used to derive sentence embeddings using a *bag-of-words* approach for sentence evaluation tasks as in the original approach.

We developed our models according to the used objective functions. Accordingly, the distance minimization method uses an encoder architecture. Whereas the second model checks the correctness of context information in a sentence since it then has to use cross-entropy loss for minimizing the perplexity; hence, it uses a decoder architecture. Additionally, in order to obtain more fine-grained decoding, an encoder architecture is added in front of the decoder structure, obtaining a sequence to sequence model.

### 3.1. Distance Minimization

The objective of this method is to reduce the disparity between sentence embeddings in different vector spaces. To achieve this, a transformer encoder architecture is being used. The generated sentence vectors are then minimized using regression. We explore the combination of two types of distance metrics as loss functions: Euclidean distance and cosine similarity. The combined loss function will help to capture the important information which the other distance metric misses.

In this approach, the *FastText* word embeddings are updated by using the outcome vectors of the *sBert* model for corresponding sentences. This enables us to augment the informational value in *FastText* embeddings, resulting in a closer resemblance to the *sBert* embeddings within the desired space.

The proposed model architecture (Figure 2) consists of a word-embedding layer and a transformer encoder structure. The encoder structure generates embeddings for all words in a sentence. For each batch, the encoder produces a tensor of shape $DxLxB$, where $D$ represents dimensionality, $L$ indicates sentence length, and $B$ denotes the batch size. The average of the tensors with shape $DxLxB$, resulting in an output tensor of shape $DxB$. Subsequently, the loss is calculated by using the Euclidean and the cosine distances, each with equal weights.

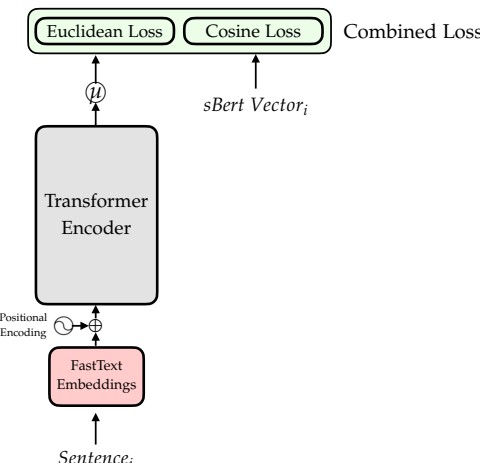

**Figure 2.** The *Distance Minimization* model consists of a word-embedding layer for updating *FastText* word embeddings, a transformer encoder structure which projects the outputs of the encoder, and combined loss for comparing the distance with *sBert* sentence vectors.

### 3.2. Distance and Perplexity Minimization

Knowledge distillation can be seen as a problem of transferring information from a teacher model to a student model, aiming to align their levels of entropy. From the perspective of natural language processing, this alignment can also be understood in terms of perplexity. Perplexity is used to measure how well a probabilistic model predicts a given sample. In our case, *sBert* demonstrates superior representational power compared to *FastText*, which enhances its robustness in prediction accuracy.

To facilitate the information transfer, we propose utilizing a sequence-to-sequence transformer model (Figure 3). In this method, the word embedding layers are shared between the encoder and the decoder parts of the model. The input sentences are initially processed by the encoder, during which a set of representations are generated for each word based on their positions in the sentences. These representations are then provided to the cross-attention module within the decoder. For each position in the target sequence, the decoder leverages the cross-attention module to interpret the relationship between the encoder's representations (key–value pairs) and its own queries to predict the subsequent word in the output sequence, utlizing cross-entropy for training, hence minimizing the perplexity values. After the decoder generates a set of representations for each word in the output sequence, these representations are then averaged to obtain sentence vectors. Finally, all generated sentence vectors are compared with their *sBert* counterparts.

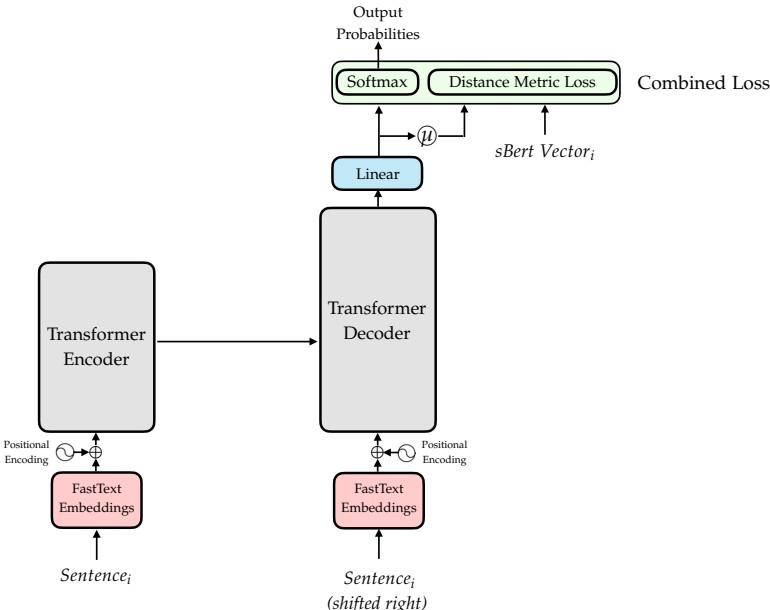

**Figure 3.** The *Distance and Perplexity Minimization* model. A sequence-to-sequence model with shared embedding layer. The embedding layer carries *FastText* word embeddings. This model aims to preserve the contextual information of words and enhance the perplexity of generated sentences for better sentence representations, which will more carefully reflect knowledge to word embedding layers.

## 4. Data

Our experiments involve three different sets of data. The first set is used specifically for information transfer purposes. The second set evaluates how the updated word embeddings are changed compared to the original ones, as well as to proposed models. Finally, the third dataset assesses how well transferred sentence embeddings perform compared to *sBert* sentence embeddings. In order to focus on a single language for our tasks, we use English datasets throughout the experiments.

### 4.1. Data for Transferring Knowledge

To facilitate the knowledge transfer from *sBert* to *FastText* sentence embeddings, the publicly accessible WMT dataset, encompassing approximately 2 million sentences, was employed. We ensured consistency and comparability by using the same randomly selected sentences for both training and testing across the proposed models. Specifically, the training subset consisted of 285 thousand randomly chosen sentences and the testing subset contained 15 thousand, culminating in a total of 300 thousand sentences. The words from the training sentences formed a vocabulary of 56 thousand words, leading to all evaluations being performed on a specifically chosen segment of the *FastText* word embeddings.

*4.2. Word-Level Evaluation: Word Analogy Tasks*

The Google Analogy Test Set [1] is a compilation of word pairs that are used to assess how well word embeddings capture types of relationships. Since our models directly work with the word embeddings, it's important to show how knowledge transfer affects the updated embeddings compared to the original ones.

The evaluation of this test set primarily focuses on two tasks; identifying syntactic and semantic relationships between word pairs. Both tasks aim to find the word for a given query based on a provided key–value pair.

*4.3. Sentence Level Evaluation*

SentEval [21] is a toolkit developed by Meta (formerly Facebook). It offers a range of tasks and linguistic probes to assess the performance of sentence embedding models. It is widely recognized as the standard tool for benchmarking datasets in sentence representation tasks.

The toolkit consists of different tasks, including classification, semantic relatedness, probing, and image caption retrieval. The classification task consists of eight datasets, including sentiment analysis, question type classification, and subjectivity or objectivity classification. The semantic relatedness tasks aims to measure the degree of similarity or dissimilarity between a pair of sentences. Since our goal is to show the relationship between embedding models, it was decided to concentrate on analyzing classification and semantic textual similarity results. The probing and image caption tasks will be excluded from our analysis.

4.3.1. Classification Tasks

The datasets listed below are intended for the development of a logistic regression classifier utilizing sentence embeddings.

- CR (Customer Review): This dataset is composed of product reviews from customers. Each piece of feedback is categorized according to the sentiment conveyed, with labels indicating positive or negative. The objective is to predict the sentiment reflected in customer product reviews.
- MR (Movie Review): This dataset features movie reviews. Each review is labeled as having either a positive or negative sentiment. The task involves predicting the sentiment of movie review.
- MPQA (Opinion Polarity): This dataset features annotations of opinion polarity within phrases from newswire texts. The challenge lies in classifying each phrase according to whether it conveys a positive or negative opinion.
- SUBJ (Subjectivity Status): This collection consists of sentences derived from movie reviews and plot summaries. The goal is to determine whether each sentence expresses personal opinions and feelings (subjective) or states factual information (objective)
- SST-2 (Stanford Sentiment Treebank): This sentiment analysis resource provides sentences extracted from movie reviews, with the objective being to determine if the sentiment is positive or negative.
- SST-5 (Stanford Sentiment Treebank): An extension of SST-2, this dataset encompasses a spectrum of sentiment annotations with five distinct levels, from very negative to very positive. It facilitates a more nuanced sentiment analysis task by allowing for fine-grained classification of emotions.
- TREC (Text REtrieval Conference): A dataset curated for the task of question classification, it differentiates between six specific types of questions, including categories like location, numeric value, and person.
- MRPC (Microsoft Research Paraphrase Corpus): is a collection of sentence pairs automatically extracted from online news sources, with annotations to indicate whether each pair captures the same semantic content, thereby serving as a resource for identifying paraphrases in parallel news articles.

4.3.2. Semantic Relatedness Tasks

SentEval is a tool that also includes semantic relatedness tasks, which are divided into two types: those evaluated by supervised models and those assessed by unsupervised models. The objective of both types is to predict a semantic similarity score between 0 and 5 for pairs of sentences, with these pairs previously scored by humans for similarity. The main challenge lies in measuring the degree to which the cosine similarity of sentence embeddings corresponds with these human judgments. The performance of both supervised and unsupervised approaches is gauged using Pearson and Spearman correlation coefficients.

Within this context, the datasets from STS12 to STS16 feature sentence pairs from a variety of sources, such as news articles, online forums, and multimedia descriptions. Adding to this, the STS Benchmark (STSB) is a curated collection of English datasets employed in the Semantic Textual Similarity (STS) tasks during SemEval workshops from 2012 to 2017, encompassing texts from image captions, news headlines, and user forums. In addition, the SICKR (Sentences Involving Compositional Knowledge for Relatedness) dataset provides a benchmark for evaluating the ability of models to understand sentence meanings in terms of semantic relatedness. Collectively, these resources form a comprehensive suite for testing computational models on text similarity tasks.

## 5. Experimental Procedure

In this section, the experimental framework is employed to validate the efficacy of our proposed methodologies. Since the information transfer is carried out between two different models, it is necessary to minimize the number of parameters to be learned. Instead of having the network learn the dimensional differences between *sBert* sentence vectors and *FastText* sentence (word) vectors (sBert sentence vectors have 768 dimensions whereas *FastText* embeddings have 300 dimensions. This situation forces the proposed models to output 300 dimensional vectors which requires an additional feed-forward network for eliminating the dimensional difference), we utilized the orthogonal mapping process to reduce the dimensions of *sBert* sentence vectors to the dimensions of *FastText* vectors. This operation requires calculation of sentence vectors from both approaches for input–output relations. Then, by calculating a simple rotation matrix, the dimensions of *sBert* are reduced to the same size of *FastText*. This process can be referred to as *supervised dimensionality reduction* [22]. We named the resulting vectors as $sBert_{Reduced}$ vectors. The test values for the *sBert* vectors obtained from this process are also given in the Section 6.

The *distance minimization* method aims to transfer knowledge by aligning the sentence embeddings generated by the encoder model with the *sBert* sentence vectors. To achieve this, the combination of Euclidean and cosine distance metrics are utilized as a loss function. The weights for each distance function are shared equally. The *distance and perplexity minimization* experiment demonstrates the effectiveness of incorporating context information to enhance the transfer of information between *sBert* sentence vectors and *FastText* word embeddings. The key difference from the previous method is the usage of a decoder substructure.

For both the *distance minimization* and *distance and perplexity minimization* methods, architectures consisting of a single layer with the following configurations were constructed: The embedding layers utilize 300-dimensional vectors sourced directly from *FastText* word embeddings. Each model employs 4 attention heads and generates 300-dimensional vectors for sentence vector comparison, analogous to *sBert* representations, which are also projected into a 300-dimensional space. During the training phase, extensive hyperparameter optimization was conducted, revealing that a learning rate of $3 \times 10^{-4}$ was optimally consistent across all models. This learning rate, in conjunction with the use of the Adam optimizer and a batch size of 128, resulted in the most effective training outcomes. All experiments are performed using an Nvidia RTX A4500 GPU.

In order to make a fair comparison, we randomly selected 300 thousand sentences from the mentioned WMT corpora. The dataset was divided into a training set comprising 285 thousand sentences and a test set containing 15 thousand sentences.

The validation results of each proposed method are given in Table 1. It is clear that the *distance and perplexity* minimization technique converges to a better cosine result.

Both proposed methods can generate sentence vectors in two ways: Either by averaging the updated *FastText* word embedding (bag-of-words approach), or by generating through the model. Although, the updated *FastText* word embeddings will be evaluated through word analogy test sets as well as sentEval evaluation sets. The model outputs will be evaluated only on sentence evaluation tasks.

**Table 1.** The validation results for *distance minimization* and *distance and perplexity minimization* methods. The *distance minimization* approach uses the combined loss of Euclidean and cosine distances. Whereas the *distance and perplexity minimization* method uses the combined loss of cosine and cross entropy. Each method uses equal weights between their loss functions.

| Model | Objective | Combined Loss | Cosine | Euclidean | XE [1] |
|---|---|---|---|---|---|
| Distance Min | EUC [2] + COS [3] | 0.38 | 0.21 | 0.58 | - |
| Distance and PP [4] Min | XE + COS | 0.08 | 0.16 | - | 0.013 |

XE [1]: Cross-Entropy, EUC [2]: Euclidean distance, COS [3]: Cosine Distance, PP [4]: Perplexity.

## 6. Results

This section outlines the impact of the proposed models on two distinct tasks: Word Similarity and SentEval. These tasks evaluate the effectiveness of the updated FastText word embeddings and the corresponding outputs from the proposed models. Our analysis focuses on a selected subset of FastText embeddings, denoted as $\text{FastText}_{subset}$ for the original. The chosen lexicon for these experiments contains 56 thousand unique words. We limit our evaluation to this subset to ensure a broad enough vocabulary for NLP tasks and to prevent result distortion from the integration of updated vectors into the existing embedding space.

For the word similarity tasks, we introduced two updated versions of *FastText* embeddings: $\text{FastText}_{updated}$ by Distance Min and $\text{FastText}_{updated}$ by Distance and PP Min, representing the enhancements made by our models.

In sentence evaluation tasks, we categorize the models as follows: $\text{sBert}_{768}$ represents the original *sBert* sentence embeddings; $\text{sBert}_{reduced}$ indicates embeddings refined through an orthogonal operation for dimension reduction; and $\text{sBert}_{384}$ serves as a comparative alternative *sBert* model. We abbreviate the *Distance Minimization* method as Distance Min, and the *Distance and Perplexity Minimization* method as Distance and PP Min. Both updated $\text{FastText}_{subset}$ versions are collectively referred to as $\text{FastText}_{updated}$. Additionally, we benchmark the execution times for each model, providing a comparison with the baseline $\text{sBert}_{768}$ and original *FastText* embeddings.

### 6.1. Word Similarity Tasks

Table 2 presents the performance of the pre-trained subset of FastText word vectors, denoted as $\text{FastText}_{subset}$, alongside the enhanced versions updated via our proposed models. The table also includes the word relationship results derived from $\text{sBert}_{768}$. For *sBert*, which utilizes sub-word tokenization, we obtained word embeddings by averaging the sub-word vectors from the model's embedding layer to construct a word vector.

Referring to Table 2, it is evident that the word embeddings updated by the *distance minimization* method unexpectedly lose all contextual information. In contrast, the *distance and perplexity minimization* method retains this context. A potential reason for this might be the underlying models: the former merely encodes sentences to compare distances with *sBert*, while the latter leverages encoded information guided by accurate predictions using *cross entropy*. Another contributing factor could be the *sBert* vectors. Since word vectors derived from the *sBert* embedding layer yield lower results, the sentence vectors from *sBert*

may distort the word embedding layers of the proposed models, altering their direction during training. Consequently, the updated embeddings underperform when compared to the original FastText$_{subset}$ word embeddings.

**Table 2.** Google Analogy Task results. sBert$_{768}$ corresponds to averaged sub-word results, FastText$_{subset}$ denotes the original embeddings, and FastText$_{updated}$ covers the embeddings enhanced by Distance Min and Distance and PP Min methods.

| Model | Semantic Accuracy | Syntactic Accuracy | Total Accuracy |
|---|---|---|---|
| sBert$_{768}$ | 30.77% | 21.83% | 22.01% |
| FastText$_{subset}$ | **98.72**% | **80.20**% | **80.58**% |
| FastText$_{updated}$ by Distance Min | 1.28% | 0.12% | 0.14% |
| FastText$_{updated}$ by Distance and PP Min | 69.87% | 52.67% | 53.02% |

*6.2. Sentence Evaluation Tasks*

This subsection shows the results of proposed approaches on the SentEval library. In order to make fair comparisons, it was decided to use the same parameters that are described on the repository page. Hence, for tasks that require classification models, batch size is set to 64, Adam is used as optimizer, number of epochs is set to 4, and each experiment is done by using a *k-fold* ($k = 10$) cross-validation method.

We also evaluate the performance of different *sBert* sentence embedding models in order to show that the used dimensionality reduction technique is capable of performing without substantial loss from its original model (sBert$_{768}$). Since our reduced *sBert* vectors (sBert$_{reduced}$) have 300 dimensions, we find it appropriate to compare it with a 384 dimensional, trained on 12-layered architecture model (sBert$_{384}$). To assess the quality of *sBert* embeddings, we utilized a script implemented using the Sentence-Transformers python package. For evaluating *FastText* sentence embeddings, we used the *bag-of-words* approach. For a detailed comparative analysis with other models in the literature, refer to [14,23], which provides extensive benchmarks across various embedding architectures by using the same datasets.

Table 3 presents the results for classification tasks. Refined *FastText* word embeddings, updated through a *distance minimization* method, underperform when compared to the original FastText$_{subset}$ across all tasks. Moreover, employing the *distance minimization* model alone does not lead to performance gains. Conversely, our second proposal, the *distance and perplexity minimization* model and its updated *FastText* embeddings demonstrate improved outcomes on *CR*, *MPQA*, *SST2*, and *MRPC* datasets in comparison to the FastText$_{subset}$. This performance improvement is partly due to the *Word Domain Coverage* metric included in the table, which assesses the overlap between the training data vocabulary used in transferring knowledge and that of the classification datasets in SentEval. Generally, datasets with higher domain coverage scores, indicating more vocabulary overlap, correspond to better results. However, the MRPC dataset represents an outlier in this trend. Unlike other datasets that primarily require the construction of a classifier, *MRPC* also demands the comparison of sentence pairs to determine semantic equivalence, which is a distinct task. Consequently, the success on *MRPC* is not solely attributable to word domain overlap but also to the model's ability to discern detailed semantic relationships. Our findings from Table 4 support this nuanced view.

**Table 3.** Comparative performance of various models on classification tasks. The *Word Domain Coverage* row indicates how the classification accuracy correlates with the overlap between the training data from the WMT dataset and the vocabulary of various classification datasets.

| Model | CR | MPQA | SST2 | MRPC | MR | SUBJ | SST5 | TREC |
|---|---|---|---|---|---|---|---|---|
| sBert$_{768}$ | 86.86 | 89.32 | 88.74 | 74.07 | 85.05 | 93.97 | 49.19 | 94.0 |
| sBert$_{reduced}$ | 85.56 | 89.18 | 88.85 | 75.42 | 84.89 | 93.45 | 48.46 | 92.0 |
| sBert$_{384}$ | 83.05 | 88.14 | 83.31 | 74.09 | 78.05 | 92.35 | 46.02 | 91.0 |
| FastText$_{subset}$ | 76.88 | 87.49 | 80.07 | 71.71 | **75.25** | **98.93** | **42.76** | **78.4** |
| FastText$_{updated}$ Distance Min | 66.31 | 83.17 | 70.95 | 67.94 | 67.15 | 81.48 | 33.98 | 66.4 |
| | 73.22 | 85.3 | 68.75 | 68.23 | 67.61 | 82.85 | 35.93 | 60.4 |
| FastText$_{updated}$ Distance and PP Min | 74.97 | **88.06** | **80.51** | 70.67 | 73.76 | 89.17 | 41.0 | 77.2 |
| | **79.42** | 87.29 | 78.36 | **72.06** | 74.13 | 88.82 | 40.9 | 76.8 |
| **Word Domain Coverage** | **0.72** | **0.86** | **0.66** | **0.59** | 0.53 | 0.52 | 0.65 | 0.60 |

**Table 4.** Results of Semantic Textual Similarity tasks on updated word embeddings and the original subset FastText word embeddings. Left values represent Pearson Correlation, whereas the right values present Spearman Correlation values.

| Model | STS12 | STS13 | STS14 | STS15 | STS16 | STSB | SICKR | Average |
|---|---|---|---|---|---|---|---|---|
| sBert$_{768}$ | 0.72 /0.70 | 0.78/0.77 | 0.80/0.77 | 0.82/0.83 | 0.82/0.83 | 0.82/0.83 | 0.87/0.82 | 0.80/0.79 |
| sBert$_{reduced}$ | 0.72/0.70 | 0.78/0.78 | 0.80/0.77 | 0.82/0.83 | 0.82/0.83 | 0.83/0.84 | 0.87/0.81 | 0.81/0.79 |
| sBert$_{384}$ | 0.71/0.70 | 0.76/0.77 | 0.80/0.77 | 0.82/0.83 | 0.81/0.82 | 0.83/0.83 | 0.87/0.80 | 0.80/0.79 |
| FastText$_{subset}$ | 0.32/0.42 | 0.39/0.41 | 0.46/0.48 | 0.53/0.55 | 0.39/0.44 | 0.61/0.61 | **0.76/0.69** | 0.49/0.51 |
| FastText$_{updated}$ Distance Min | 0.29/0.37 | 0.30/0.31 | 0.38/0.41 | 0.42/0.44 | 0.31/0.36 | 0.47/0.47 | 0.64/0.61 | 0.40/0.42 |
| | **0.47/0.48** | **0.55/0.55** | **0.54/0.51** | **0.59/0.59** | **0.54/0.56** | 0.50/0.49 | 0.59/0.53 | **0.54/0.53** |
| FastText$_{updated}$ Distance and PP Min | 0.33/0.44 | 0.39/0.41 | 0.46/0.48 | 0.52/0.54 | 0.39/0.44 | 0.57/0.58 | 0.74/0.68 | 0.49/0.51 |
| | **0.55/0.57** | **0.56/0.58** | **0.61/0.58** | **0.66/0.66** | **0.62/0.63** | **0.63/0.62** | **0.76**/0.67 | **0.63/0.62** |

Table 4 delineates the results for semantic textual similarity (STS) tasks. Mirroring the trends observed in classification tasks, embeddings updated via the *distance minimization* method continue to underperform. However, when these embeddings are further processed through the *distance minimization* model, there is a notable improvement, with performance exceeding that of the original *FastText* (FastText$_{subset}$)embeddings in five out of seven STS tasks, indicative of a superior mean performance. Interestingly, words updated through the *distance and perplexity minimization* method yield results that are comparable to the original FastText$_{subset}$ embeddings. Yet, when sentence pairs are evaluated using this model, there is a marked improvement, outperforming the FastText$_{subset}$ in six out of seven tasks.

The obtained results from both tasks suggest that the efficacy of used methods is more pronounced at the sentence level, emphasizing the importance of the chosen model structure, loss function, and its alignment with the task-specific demands.

*6.3. Runtime Comparison*

To illustrate the impact of the proposed models on sentence embedding generation, we show the execution times across various models. These models were benchmarked on an identical dataset comprising 300,000 sentences, which we used for training and validating the proposed models. All models utilized GPU acceleration, except *FastText*, and were executed in batches of 64. For obtaining sentence embeddings from the sBert$_{768}$ model, the Sentence-Transformers (https://www.sbert.net, accessed on 23 October 2023) python package was used.

Table 5 indicates that the sBert$_{768}$ model exhibits the longest duration for sentence embedding generation. In contrast, *FastText* is uniquely efficient, functioning effectively

without GPU support. The models that we propose serve as an intermediate solution, offering a compromise between these two in terms of computational resource demands and time efficiency. Moreover, our implementations can be optimized further for better execution times.

**Table 5.** Execution times for sentence representation generation across different models.

| Model | Time (s) |
|---|---|
| sBert$_{768}$ | 271.7 |
| FastText$_{subset}$ | 23.46 |
| Distance Min | 88.4 |
| Distance and PP Min | 182.67 |

## 7. Discussion

This paper introduced alternative strategies for scenarios where traditional knowledge distillation is impractical due to inaccessible model parameters or incompatibility across different deep learning frameworks. We have explored novel methods that circumvent these constraints, focusing on enhancing sentence vector alignment using *FastText* embeddings, diverging from the established norms of *sBert* sentence vectors and knowledge distillation techniques.

Unlike *sBert*, which employs contrastive learning to cluster similar sentences together and separate dissimilar ones, our approach is built on the foundation of sentence vector training with a unique twist. We utilize vectors derived from disparate architectures and employ loss functions that concentrate on sentence similarities, eschewing the contrasting aspect of *sBert*.

From a knowledge distillation perspective, standard methodologies typically focus on aligning hidden states or reproducing logits between models. Our innovative approach diverges by utilizing the average of pre-logit hidden states, which facilitates the distillation process using just a single vector, as opposed to employing the entirety of pre-logit hidden state vectors.

Our experiments with dual combined loss functions yielded mixed outcomes. The *distance minimization* strategy, which adjusts *FastText* embeddings via an encoder model, unfortunately led to the degradation of word content, resulting in subpar sentence vector production. In contrast, the *distance and perplexity minimization* model utilizes a more nuanced loss function that combines cross-entropy, for capturing the distribution of words, with cosine similarity, for aligning vectors. This approach highlights the importance of complex loss functions and domain relevance in creating robust models capable of generalizing across diverse linguistic contexts. Furthermore, our in-depth analysis revealed that datasets with a higher domain coverage, denoting more shared vocabulary with the training data, are associated with enhanced performance in classification tasks. This observation reinforces the promise of our proposed models in computational linguistics applications.

Building on these insights, our research was primarily aimed at information transfer between different sentence embedding models. Here, we discovered an interesting phenomenon: reducing the dimensions of the original *sBert* model does not affect the sentence evaluation tasks. This stands in contrast to the common assumption that higher dimensions are synonymous with better performance; our findings show that a pared-down model can achieve comparable results. This revelation adds a new layer to our understanding of model efficiency and underscores the potential of dimensionality optimization in natural language processing tasks.

## 8. Conclusions

As we reflect on the validation phase of our research, the potential for improvement in capturing the full spectrum of information from the sBert embedding space has been

significant. These insights set the stage for future research to refine calibration techniques within representation spaces, potentially through the expansion of training corpora size.

Our aspiration extends beyond addressing the intricacies of knowledge transfer between diverse architectures; we aim to set new standards for model interoperability across various deep learning frameworks where all model parameters are not shared publicly. The challenges faced in this pursuit, including data scarcity and model complexity, will inform our forward trajectory, guiding the development of more adaptable and user-friendly tools. By overcoming these obstacles, we can advance the field, unlocking the full potential of sophisticated sentence embedding models for a wider range of applications.

**Author Contributions:** Conceptualization, K.G. and M.F.A.; Methodology, K.G. and M.F.A.; Software, K.G.; Validation, K.G. and M.F.A.; Formal analysis, K.G. and M.F.A.; Investigation, K.G. and M.F.A.; Resources, K.G. and M.F.A.; Data curation, K.G. and M.F.A.; Writing—original draft, K.G.; Writing— review & editing, K.G. and M.F.A.; Visualization, K.G.; Supervision, M.F.A.; Project administration, M.F.A. All authors have read and agreed to the published version of the manuscript.

**Funding:** This research received no external funding.

**Institutional Review Board Statement:** Not applicable.

**Informed Consent Statement:** Not applicable.

**Data Availability Statement:** The data used for training proposed methods can be found in http://www2.statmt.org/wmt23/translation-task.html#_data (access on 23 October 2023). Google's Analaogy Tasks can be found in http://download.tensorflow.org/data/questions-words.txt (access on 23 October 2023). For evaluating the proposed methods, the sentence evaluation tasks can be found in https://github.com/facebookresearch/SentEval (access on 23 October 2023). The *FastText* word embeddings used during training of proposed methods and creating sentence embeddings can be downloaded from https://dl.fbaipublicfiles.com/fasttext/vectors-english/crawl-300d-2M.vec.zip (access on 23 October 2023). The *sBert* models used during knowledge transfer and for comparison purposes can be downloaded from *HuggingFace*: https://huggingface.co/sentence-transformers/all-mpnet-base-v2/resolve/main/pytorch_model.bin?download=true (access on 23 October 2023) and https://huggingface.co/sentence-transformers/all-MiniLM-L12-v2/resolve/main/pytorch_model.bin?download=true (access on 23 October 2023).

**Conflicts of Interest:** The authors declare no conflict of interest.

## Abbreviations

The following abbreviations are used in this manuscript:

| | |
|---|---|
| XE | Cross Entropy |
| PP | Perplexity |
| EUC | Euclidean |
| COS | Cosine |

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
