# Peer review of "Boosting Lightweight Sentence Embeddings with Knowledge Transfer from Advanced Models: A Model-Agnostic Approach"

_applsci, doi:10.3390/app132312586_

Round 1
Reviewer 1 Report
Comments and Suggestions for Authors
The study explores transferring knowledge from a powerful, computation-intensive sentence embedding model to a simpler model that averages word vectors. Instead of conventional methods, they use the output vectors of the advanced model to align with the simpler model's word vectors. Two techniques, distance minimization and distance & perplexity minimization, are applied. Training uses the WMT datasets, while validation is done with Google’s Analogy tasks and Meta’s SentEval datasets. The results show that the word embeddings incorporate information in a unique, model-specific way.
The work presents an innovative approach to knowledge transfer in the realm of sentence embedding models. It's commendable that the authors are trying to leverage the strengths of a powerful model to improve a lightweight model, which could have real-world implications for applications where computational efficiency is vital. The choice to deviate from traditional distillation methods and to focus on output vectors is intriguing, as it can potentially simplify the transfer process.
The use of well-established datasets like WMT for training and Google’s Analogy tasks and Meta’s SentEval for validation lends credibility to the research. However, the conclusion that embeddings acquire information in a "distorted, model-specific fashion" raises questions. This statement suggests there could be limitations in how generalizable or interpretable the transferred knowledge is.
I thought the work was engaging and thought it deserved to be published. However, I have some suggestions for the authors to consider.
Comments
Suggestion of title: Boosting Lightweight Sentence Embeddings with Knowledge Transfer from Advanced Models: A Model-Agnostic Approach
In the abstract, it would be beneficial to have more insights or explanations about what "distorted, model-specific fashion" means in practical terms and the implications of such a distortion.
Regarding the Introduction, it provides a concise overview of knowledge distillation, explaining the relationship between teacher and student models, which helps set the stage for the paper's main focus. The introduction points out the challenges and limitations of traditional knowledge distillation techniques, giving readers a clear reason for the study's importance. The paper's primary focus on knowledge transfer between FastText and sBert is well-articulated, and the distinct characteristics of these models are highlighted. The purpose of the study, which is to bridge the representational gap between FastText word embeddings and sBert sentence vectors, is clearly mentioned. The organization of the paper is well-outlined, guiding readers on what to expect in the upcoming sections.
However, there's some redundancy in the text. For instance, the concept of the student model trying to mimic or acquire representations from the teacher model is repeated multiple times. Some sentences are awkwardly phrased or have minor grammatical errors. E.g., "be lack of prediction capabilities" should be "lack prediction capabilities." The introduction mentions scenarios where distillation methods can be challenging due to limited access to model parameters or the need to redevelop models. However, the significance of these challenges and how they relate to the paper's primary focus isn't elaborated upon. The statement that "generating sentence vectors with FastText is very efficient and calculations rely on CPU" might need further explanation. Some might interpret that FastText exclusively uses CPU, which may not be accurate. The introduction states that sBert excels over FastText in terms of accuracy and similarity tests. However, this is introduced suddenly without a clear lead-in. The transition between discussing the models' technical specifications and their performance could be smoother.
Regarding the Literature review, it provides a chronological progression of developments in sentence vector representations, making it easy to follow the evolution of the field. It touches upon various techniques ranging from simple averaging of word embeddings to more sophisticated models like BERT, giving readers a broad perspective on the topic. While it covers various models and methods, it delves into specific characteristics and contributions of certain important ones. The review ends by differentiating the current study from existing literature, which helps in understanding the paper's unique contribution.
However, some transitions between topics or papers could be smoother. For example, "Apart from sentence generation model, knowledge distillation is a widely explored technique..." seems abrupt. Some methods or papers are explained in detail while others just get a passing mention. A bit more consistency would be beneficial. There are a few grammatical errors and awkward phrasings, such as "Models such as Bert[11] opened new opportunities for which can be exploited for..."
Use smoother transitions between various topics to ensure the literature review flows seamlessly. The review ends by stating the unique focus of the current study, but it might be helpful to underscore this difference more prominently. For seminal works or models like BERT, consider providing a slightly more detailed explanation given their significance in the field. Briefly explain why each cited work is relevant to the current study. This can help readers connect the dots more easily.
Regarding the Results section, it is comprehensive with evaluations across different tasks, including word similarity and sentence evaluations. The tables present a clear comparison of the original embeddings, updated embeddings, and various models, allowing readers to quickly grasp the performance of each approach. The reasons for evaluating only a subset of word embedding space and the experimental setups for SentEval library are provided, which increases the transparency and reproducibility of the experiments.
However, the information, especially around the reasons for evaluating only a subset and the underlying models, is dense. Some readers might find it difficult to quickly grasp the significance of specific results. The term 'FastTextupdated' appears twice in Table 3, which can be confusing. It’s unclear if they represent different models or if this is a typographical error. There's a sudden mention of "Distance Min.," which seems to refer to "Distance Minimization." While abbreviating can save space, it can sometimes cause confusion if not introduced properly. While tables are effective, visual representations like bar graphs or line plots might help readers visualize the performance differences more effectively, especially when there are multiple models or methods to compare. Although the section does compare updated embeddings against original embeddings, it might benefit from a comparison with standard or widely-accepted benchmarks in the field. There's a mention of sBert’s word relations and sBert vectors, but without proper context, it's not immediately clear to unfamiliar readers what role these vectors play. Some of the explanations, like the description around "Distance Minimization" and "Distance & Perplexity Minimization," could be streamlined for clarity.
Resolve the repeated use of 'FastTextupdated' in Table 3 to ensure clarity. Consider including visual aids like graphs to represent performance differences more vividly. After presenting the tables, a brief summary of the main takeaways will help the reader consolidate their understanding. Introduce abbreviations and acronyms with care, ensuring that they don't add confusion. If possible, compare the results with well-established benchmarks in the field to provide context about the performance of the proposed models. Streamlining the language and breaking down complex sentences can enhance clarity and readability. Consider diving deeper into the reasons behind certain anomalies in results (e.g., why the Distance Minimization method loses all word context information).
The Discussion section effectively begins by establishing the environment in which their study functions, emphasizing the importance of their approaches in instances when standard information distillation is not practicable. The distinctions between the usual approach and the proposed methods are well outlined, allowing readers to readily grasp the study's new characteristics. The discussion delves further into both techniques, giving readers insight into why certain strategies were more beneficial than others. It finishes with a clear direction for future research, referring to calibration of representation spaces as a promising path for further investigation.
However, the discussion, while detailing the approach and results, lacks an in-depth comparative analysis with other existing methods or similar studies. Statements such as "it can also be assumed that the updated word embeddings converge to sBert’s word embedding layers" need more justification or evidence.
Sentences like "Our unique proposition entails utilizing the average of pre-logit hidden states which makes the distillation process only by using a single vector" could benefit from clarification. While differences between the conventional and the proposed methods are highlighted, more nuanced differences, pitfalls, or challenges faced could be discussed to provide a comprehensive view of the study's journey.
Moreover, investigate potential real-world applications or use-cases that could benefit from the improved embedding strategy. Compare and contrast the computational costs, time difficulties, and resource consumption of the proposed methods to those of traditional methods. Address the difficulties encountered during the study in greater depth, possibly expanding on how they were (or were not) solved. Consider how the procedures could be extended beyond the specific models mentioned. Can the method be applied to other models or frameworks? Discuss whether or not any hyperparameter tuning or optimization was undertaken, and if not, speculate on the potential consequences. Investigate whether the results have implications or applications in other fields or interdisciplinary areas.
There is no section about the Conclusion.
The paper’s abbreviations are defined near the end. Please incorporate them into the main text.
Author Response
Please see the attachment.
Kindly Regards,
Kadir Gunel

Reviewer 2 Report
Comments and Suggestions for Authors
General Comment:
The paper investigates knowledge transfer between distinct sentence embedding models, specifically between a computationally demanding yet high-performing model and a lightweight model derived from word vector averaging. The primary objective is to augment the representational power of the lightweight model by incorporating features from the robust model. Authors aim to explore an alternative approach to knowledge distillation, focusing on aligning the output sentence vectors of the teacher model with the student model's word vector representations. The implementation of two minimization techniques, distance minimization and distance & perplexity minimization, is a notable aspect of this research. The methodology employs WMT datasets for training, and the resulting enhanced embeddings are assessed using Google's Analogy tasks and Meta's SentEval datasets.
Comment #1 - Literature Review:
The manuscript's literature review section is notably lacking in depth and breadth. With less than a page of content and only 20 references, it does not provide sufficient context or demonstrate a comprehensive understanding of the existing body of knowledge. To strengthen the research, the authors should expand the literature review. Incorporate more recent references to reflect the current state of research in the field, including key and recent publications. The addition of recent work, such as Leone et al. (2020), Audrito et al. (2023), and Pan et al. (2019), is recommended to enrich the literature review.
• Leone, V., Siragusa, G., Di Caro, L., & Navigli, R. (2020, May). Building semantic grams of human knowledge. In Proceedings of the Twelfth Language Resources and Evaluation Conference (pp. 2991-3000).
• Audrito, D., Sulis, E., Humphreys, L., & Di Caro, L. (2023). Analogical lightweight ontology of EU criminal procedural rights in judicial cooperation. Artificial Intelligence and Law, 31(3), 629-652.
• Pan, C., Huang, J., Gong, J., & Yuan, X. (2019). Few-shot transfer learning for text classification with lightweight word embedding based models. IEEE Access, 7, 53296-53304.
Comment #2 - Conclusion:
The absence of a Conclusion section is a notable gap in the manuscript. Authors should incorporate a well-developed conclusion to summarize the primary findings of the study. Authors need to discuss the implications of the findings, underscore the personal contributions of the research, and suggest future research directions. Additionally, the practical implications of the study and its contribution to the field should be reflected upon. Also, a discussion of the study's limitations to provide a balanced perspective, is needed.
Final Comment:
The manuscript presents a valuable study on knowledge transfer between sentence embedding models, and may be considered for publishing in „Applied Sciences”, Special Issue: „Natural Language Processing: Novel Methods and Applications”. However, to enhance its quality and completeness, it is mandatory that the authors prioritize expanding the Literature review and add a well-structured Conclusion section.
Comments on the Quality of English LanguageThe manuscript demonstrates a reasonable command of the English language. However, some moderate editing is needed to enhance overall readability, including addressing grammatical issues. The paper's overall structure is coherent and logical, but it could benefit from improved, clear transitions between sections that would facilitate a smoother flow.
Author Response

(The authors gave the same response as above.)

Round 2
Reviewer 2 Report
Comments and Suggestions for Authors
Dear Authors,
You have commendably addressed the previous comments and substantially strengthened the manuscript. The revisions reflect a thoughtful consideration of (almost all of) the feedback provided, enhancing the overall quality and coherence of the paper.
Good-luck with your manuscript and wish you all the best!
Warm regards,
Reviewer